

# Malformed trilobites from the Cambrian, Ordovician, and Silurian of Australia

Russell D. C. Bicknell[1,2], Patrick M. Smith[3,4] and John R. Paterson[2]

[1] Division of Paleontology (Invertebrates), American Museum of Natural History, New York, NY
[2] Palaeoscience Research Centre, School of Environmental and Rural Science, University of New England, Armidale, New South Wales, Australia
[3] Palaeontology Department, Australian Museum Research Institute, Sydney, New South Wales, Australia
[4] Department of Biological Sciences, Macquarie University, Sydney, New South Wales, Australia

## ABSTRACT

Biomineralised remains of trilobites provide important insight into the evolutionary history of a diverse, extinct group of arthropods. Their exoskeletons are also ideal for recording malformations, including evidence of post-injury repair. Re-examination of historic collections and the study of new specimens is important for enhancing knowledge on trilobite malformations across this diverse clade. To expand the records of these abnormalities and present explanations for their formation, we document eight malformed trilobite specimens, as well as one carcass, housed within the Commonwealth Palaeontological Collection at Geoscience Australia in Canberra. We present examples of *Asthenopsis*, *Burminresia*, *Centropleura*, *Coronocephalus*, *Dolicholeptus*, *Galahetes*, *Papyriaspis*, and *Xystridura* from Cambrian, Ordovician, and Silurian deposits of Australia. Most of the malformed specimens show W-, U-, or L-shaped indentations that reflect injuries from either failed predation or complications during moulting, and a mangled carcass is ascribed to either successful predation or post-mortem scavenging. We also uncover examples of teratologies, such as bifurcated pygidial ribs and pygidial asymmetry, in addition to evidence of abnormal recovery (*i.e.*, fusion of thoracic segments) from a traumatic incident.

## INTRODUCTION

Specimens of extinct animals displaying malformations present important and often unique insight into predation, pathological growths, and genetic abnormalities in the fossil record (*Owen, 1985*; *Babcock, 1993*, *2003*, *2007*; *Kelley, Kowalewski & Hansen, 2003*; *Huntley, 2007*; *Klompmaker & Boxshall, 2015*; *Leung, 2017*). Malformations have been documented in many fossil groups (*Klompmaker et al., 2019*) and are especially well known from the extinct group of arthropods called trilobites (*Šnajdr, 1978*; *Owen, 1983*, *1985*; *Jell, 1989*; *Babcock, 1993*, *2003*; *Fatka, Budil & Grigar, 2015*; *Fatka, Budil & Zicha, 2021*; *Bicknell, Paterson & Hopkins, 2019*; *Bicknell & Holland, 2020*; *Zong, 2021b*; *Bicknell & Smith, 2022*; *De Baets et al., 2022*). The extensive record of malformed trilobites is facilitated by their biomineralised exoskeleton (*Babcock, 1993*, *2003*) that increased the preservational potential of specimens and, by extension, the ability to record abnormal

Corresponding author
Russell D. C. Bicknell,
rdcbicknell@gmail.com

structures (*Bicknell et al., 2022c*). Thus, trilobites represent an ideal group for understanding malformations in extinct arthropods.

Geoscience Australia in Canberra hosts one of the largest reference collections of Australian fossil material and is home to the extensive Commonwealth Palaeontological Collection (CPC) (*Schroeder & Laurie, 2023*). Of particular note are the specimens documented in the seminal works of Armin Öpik, John Shergold, and Desmond Strusz that have major scientific and historical importance for our understanding of Australian trilobites and their use in biostratigraphy. Some of these key publications include illustrations of malformed trilobites (*Öpik, 1961*, *1975*, *1982*; *Strusz, 1980*), prompting us to revisit the Geoscience Australia collections (including the CPC) to see if new information can be uncovered. In doing so, we present eight malformed trilobite specimens and one possible carcass from Australian fossil deposits ranging in age from the Cambrian (Miaolingian, Wuliuan) to the Silurian (Wenlock, Sheinwoodian to Homerian). These novel and under-examined records of Australian trilobites with injuries and abnormalities present new insight into the palaeobiology of these extinct arthropods.

## MATERIAL AND METHODS

Trilobite material from Geoscience Australia (Canberra) was examined for malformations, including previously published specimens in the CPC (*Öpik, 1961*, *1975*, *1982*; *Strusz, 1980*) and unpublished material housed within the bulk collections. Specimens documented herein are from the Cambrian Arthur Creek and Beetle Creek formations and Devoncourt and V-Creek limestones in the Georgina Basin (Queensland and Northern Territory), the Ordovician Emanuel Formation in the Canning Basin (Western Australia), and the Silurian Walker Volcanics in the Lachlan Orogeny (Australian Capital Territory); detailed locality and stratigraphic information is provided below. Specimens were coated in ammonium chloride sublimate and photographed under low angle LED light with a Canon EOS 5Ds using MP-E 65 mm 1–5× macro and 50 mm lenses at the University of New England (UNE), Armidale, Australia. Images were stacked using Helicon Focus 7 (Helicon Soft Limited, Kharkiv, Ukraine) stacking software. Images were converted to greyscale after photography.

### Geological context

**Beetle Creek Formation**—Holotype CPC 10348 of *Galahetes fulcrosus Öpik, 1975* (Figs. 1A and 1B) was collected from a unit assigned to the Beetle Creek Formation in the southern part of the Burke River Outlier near Galah Creek at locality "D135" (approximately 21°57′ S, 139°36′ E) in the Georgina Basin, western Queensland (*Öpik, 1975*). Here, the unit consists of several hundred metres of siliceous shale and chert with interbedded lenses of bituminous limestone forming low cuestas and mesas in the landscape (*Carter & Öpik, 1963*). These fine-grained sediments, in addition to the presence of oryctocephalid trilobites and agnostids (including complete exoskeletons), all suggest that the unit was likely deposited below storm wave base at the outer shelf edge of a large epeiric sea (*Fleming, 1977*; *Kruse, 2002*). The unit at locality "D135" unconformably overlies the Thorntonia Limestone and is overlain conformably by the Inca Formation

(*Carter & Öpik, 1963*; *Southgate & Shergold, 1991*; *Dunster et al., 2007*). The occurrence of *Galahetes fulcrosus* in the NTGS Elk 3 and Baldwin one drill cores of the southern Georgina Basin (*Laurie, 2004, 2006a*) suggest the taxon ranges from the *Pentagnostus krusei* Zone to the *Pentagnostus praecurrens* Zone. At locality "D135", co-occurrence of both *Oryctocephalites* cf. *gelasinus Shergold, 1969* and *Sandoveria lobata Shergold, 1969* suggests an age within the upper portion of this range. Both associated taxa are found at locality "N32" in the Northern Territory with *Pentagnostus praecurrens* (*Westergård, 1936*) (= "*Pentagnostus rallus*" of *Öpik, 1979*; see *Laurie, 2004*). *Sandoveria lobata* is also known to co-occur with *Pentagnostus praecurrens* (*Westergård, 1936*) in the informal 'White Shale' member of the Coonigan Formation (*Shergold, 1969*; *Smith et al., 2023*) in the Mutawintji Ranges, western New South Wales (= "*Pentagnostus veles*" of *Öpik, 1979*; see *Laurie, 2004*). The Australian *P. praecurrens* Zone is equivalent to the lower Templetonian Stage in Australia, which partly equates to the Wuliuan Stage in the global Cambrian timescale (*Sundberg et al., 2016*; *Peng, Babcock & Ahlberg, 2020* and references therein).

**Arthur Creek Formation**—*Xystridura altera Öpik, 1975* specimen CPC 10407 (Figs. 1C and 1D)—previously illustrated by *Öpik (1975*, pl. 30, fig. 7) as *Xystridura remorata Öpik, 1975* (see *Laurie, 2006a*)—was collected from float at locality "N37" in the Sandover River, near Argadargada Station, Northern Territory, which is attributed to the 'Sandover Beds' (now part of the Arthur Creek Formation, *sensu Stidolph et al., 1988*) in the Georgina Basin. Since the specimen was found in river gravels, it is difficult to assess the original depositional environment. However, more generally, the Arthur Creek Formation (= 'Sandover Beds') in the Elkedra 200,000 km map sheet area represents a restricted, subtidal environment just above wave base (*Stidolph et al., 1988*). This is similar to the Beetle Creek Formation, although likely more proximal to a shelf edge of the large epeiric sea (*Dunster et al., 2007*). *Xystridura altera* is known to co-occur with *Pagetia prolata Jell, 1975* in the Wonarah Formation at site "N25B", 5 km northwest from Alexandria Homestead, Northern Territory. *Pagetia prolata* also occurs in the Arthur Creek Formation in the Baldwin 1 core, which contains assemblages that range from the early Templetonian *Pentagnostus anabarensis* Zone to the Floran *Euagnostus opimus* Zone (*Laurie, 2006a, 2012*). A similar age range is observed for *X. altera* in the Jigaimara Formation on Howard Island, Northern Territory (*Laurie, 2006b*). Here, *X. altera* co-occurs with *Itagnostus* sp., which is similar to (and likely synonymous with) '*Peronopsis*' *normata* (*Whitehouse, 1936*). Hence, *X. altera* ranges from the *P. praecurrens* Zone (based on the Beetle Creek Formation type section) to the *Acidusus atavus* Zone. Therefore, in combining these ranges, the "N37" material has an age somewhere between the *Pentagnostus shergoldi* Zone and *Acidusus atavus* Zone. This is equivalent to the late Templetonian and early Floran in the Australian Cambrian Stage scheme, or Wuliuan to Drumian stages in the global timescale (*Peng, Babcock & Ahlberg, 2020*) and references therein.

**V-Creek Limestone**—Paratype CPC 18907 of *Dolicholeptus licticallis Öpik, 1982* (Fig. 2E) was collected from locality "M41" in the lower V-Creek Limestone within a creek channel along Old Burketown Road (approximately 19°27.5′ S, 138°37′ E), Georgina Basin, Queensland. Likewise, the paratype CPC 18897 of *Dolicholeptus ansatus Öpik, 1982* (Fig. 2D) was collected from the same unit at locality "M54" in Douglas Creek, south of

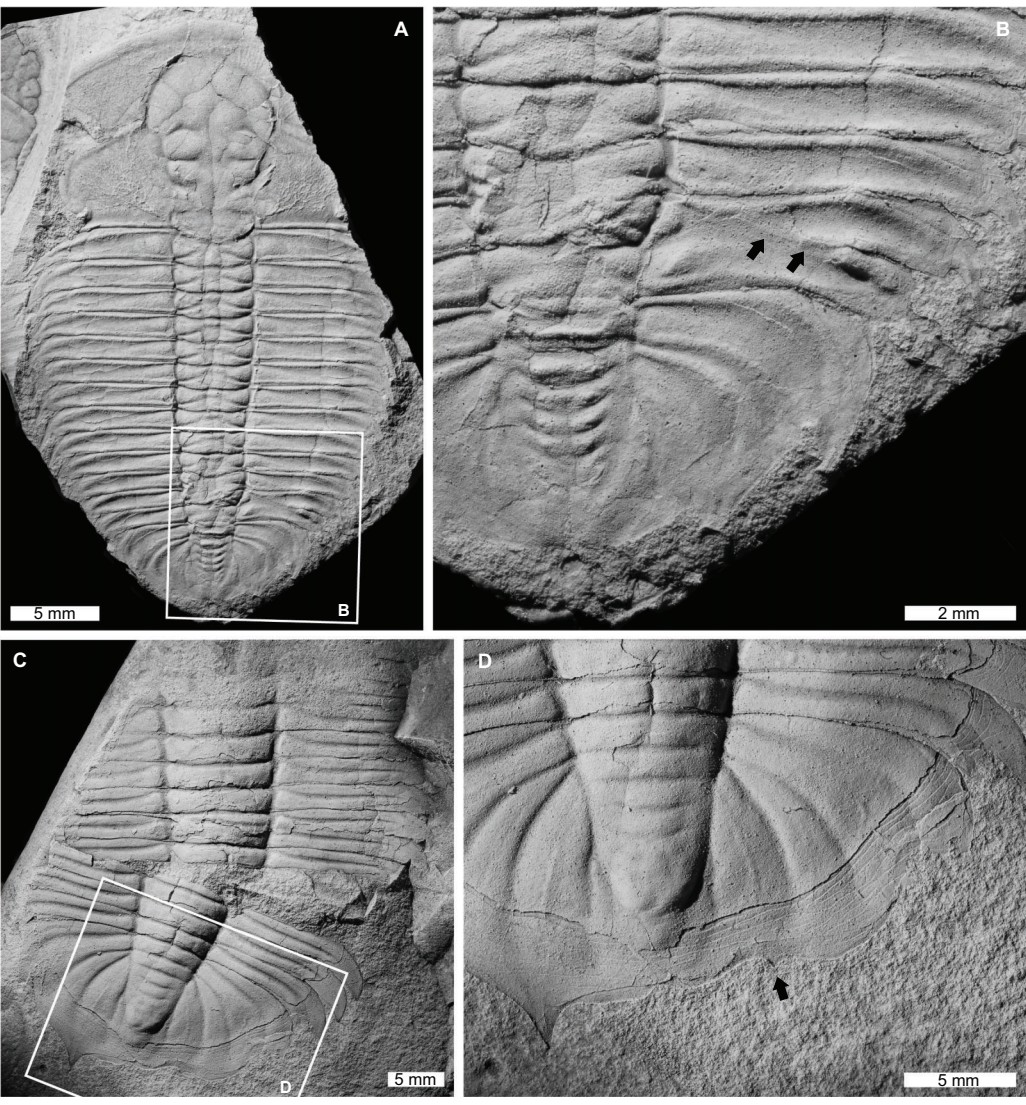

**Figure 1 Injured xystridurine trilobites from the Cambrian of Australia.** (A and B) *Galahetes fulcrosus Öpik, 1975*, CPC 10348, holotype, from the Cambrian (Miaolingian, Wuliuan) Beetle Creek Formation, Queensland. (A) Entire specimen. (B) Close up of malformed region showing fused thoracic pleurae (on T12 and T13) and axial rings (on T11 and T12); black arrows indicate the poorly defined anterior boundary of the 13th pleura. (C and D) *Xystridura altera Öpik, 1975*, CPC 10407, Cambrian (Miaolingian, Wuliuan to Drumian) Arthur Creek Formation, Northern Territory. (C) Entire specimen. (D) Close up of W-shaped indentation on right side, with evidence of spine recovery (black arrow).

Undilla Homestead (approximately 19°37′ S, 138°38′ E), Queensland (*Öpik, 1979*). An unpublished specimen of *Papyriaspis lanceola Whitehouse, 1939* (CPC 44539; Figs. 2A and 2B) was also collected from the V-Creek Limestone at locality "M418", ~3.5 km southeast of Douglas Creek, next to an unnamed dirt track (approximately 19°27.5′ S, 138°33′ E), Queensland. Finally, an unpublished specimen of *Asthenopsis* sp. (CPC 44540; Fig. 2C) from the V-Creek Limestone was found in the bulk collections at Geoscience Australia, but was not associated with locality information; lithological characteristics of

the matrix suggest that the specimen may be from (or near) locality "M418". The lithologies of all localities of the V-Creek Limestone are remarkably consistent, comprising primarily of irregular, nodular grey marly limestone (*Öpik, 1979*) that is typically conformably overlain by the Inca Formation (*Dunster et al., 2007*). Presence of finely laminated sediments, common sponge spicules, as well as a diverse assemblage of agnostids and trilobites, of which many specimens are articulated (*Öpik, Carter & Noakes, 1959*; *Öpik, 1970a*, *1979*, *1982*; *Jell, 1978*; *Paterson, 2005*), suggest that the unit was likely deposited in a quiescent, shallow marine environment (*Henderson & Dann, 2010*). Locality "M41" contains the co-occurring agnostid *Ptychagnostus punctuosus* (*Angelin, 1851*), firmly placing it in the lower Undillan stage of the Australian scheme. Locality "M54" contains the agnostids *Doryagnostus incertus* (*Brøgger, 1878*) and *Hypagnostus parvifrons* (*Linnarsson, 1869*; *Öpik, 1979*), suggesting an age somewhere between the *Ptychagnostus punctuosus* to *Goniagnostus nathorsti* zones (*Peng & Robison, 2000*). The presence of *Dolicholeptus ansatus* Öpik, 1982 in the "*Amphoton* Band" of the Knowsley East Formation, Victoria, along with *Hypagnostus parvifrons* (*Linnarsson, 1869*) and *Nepea nans* (*Öpik, 1970a*), indicates a potentially older *Euagnostus opimus* Zone age (*Jell, 2014*). Occurrence of *Goniagnostus nathorsti* (*Brøgger, 1878*) at locality "M418" indicates an age within the eponymous zone of the upper Undillan stage. In summary, all three V-Creek Limestone localities generally fall within the range of the Undillan that is equivalent to the upper Drumian in the global timescale (*Peng, Babcock & Ahlberg, 2020* and references therein).

**Devoncourt Limestone**—Paratype CPC 3494 of *Centropleura phoenix Öpik, 1961* (Figs. 3A and 3B) was collected from site "D16", which *Öpik (1961)* placed in the lower part of the Devoncourt Limestone in the Georgina Basin. The specimen is from a shallow pit near small hills of limestone along the Cloncurry-Duchess Road (approximately 21°20′ S, 140°03′ E) in the Selwyn Range, Burke River area, western Queensland (*Öpik, 1961*). Here the outcrop is a grey, sandy, flaggy foetid limestone; it is conformably underlain by the Roaring Siltstone and conformably overlain by the Selwyn Range Limestone (*Öpik, 1961*). The sedimentological and palaeontological characteristics of the unit (see *Dunster et al., 2007*) suggest an intertidal to shallow subtidal marine ramp (*Shergold & Druce, 1980*; *Southgate & Shergold, 1991*), with occasional dysoxia (*Shergold et al., 1976*). Co-occurrence of *C. phoenix* and *Lejopyge laevigata* (*Dalman, 1828*) at the "D7/15" and "D13A" sites within the Devoncourt Limestone, suggest an age within the eponymous zone of the latter taxon. Following the Australian Stage scheme, this would place the material in the lower Boomerangian, which is equivalent to the lower Guzhangian in the global timescale (*Peng, Babcock & Ahlberg, 2020* and references therein).

**Emanuel Formation**—Paratype CPC 31981 of *Burminresia prima Laurie & Shergold, 1996b* (Figs. 3C and 3D) was collected from horizon 705/131 in the Emanuel Formation at its type section along Emanuel Creek (approximately 18°39′9.8″ S 125°54′29.1″ E), Lennard Shelf, Canning Basin, Western Australia (*Laurie & Shergold, 1996a*). Here the unit is 435 m thick and consists of three informal members: (1) a poorly exposed basal member (142 m) of thin beds of limestone, shale and siltstone; (2) a middle member (143 m) with prominent light grey limestone beds, plus green-grey siltstone and shale

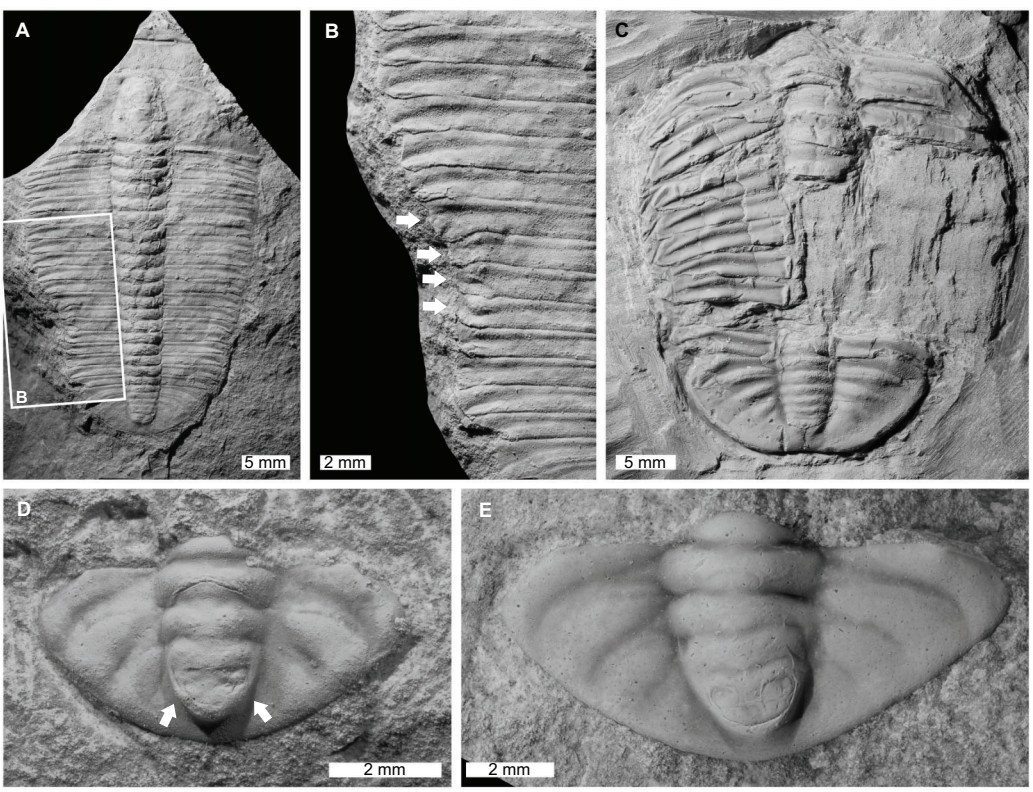

**Figure 2 Malformed trilobites and a carcass from the Cambrian (Miaolingian, Drumian) V-Creek Limestone, Queensland.** (A and B) *Papyriaspis lanceola Whitehouse, 1939*, CPC 44539. (A) Entire specimen. (B) Close up of malformed region showing L-shaped injury and deformed pleural tips (white arrows). (C) *Asthenopsis* sp. CPC 44540. Possible carcass showing removal of large thoracic regions and displacement of broken pleurae on the left side of the thorax, likely the result of predation or post-mortem scavenging. (D) *Dolicholeptus ansatus Öpik, 1982*, CPC 18897, paratype. Pygidium showing partial collapse of the axis (white arrows), which *Öpik (1982)* previously interpreted as an abnormality. (E) *Dolicholeptus licticallis Öpik, 1982*, CPC 18907, paratype. Asymmetrical pygidium showing the axis deflected to the right side.

containing interbedded limestone nodules; and (3) a poorly exposed upper member (150 m) with shale, and siltstone interbedded with smaller limestone nodules which decrease in abundance up section (*Nicoll, Laurie & Roche, 1993*; *Shergold, Laurie & Nicoll, 1995*; *Laurie & Shergold, 1996a*). The unit overlies the Kudata Dolostone and is conformably overlain by the Gap Creek Formation in nearby Gap Creek (*Laurie & Shergold, 1996a*). It has been suggested that the Emanuel Formation was likely deposited in a relatively deep water, mid-outer shelf environment (*Zhen & Nicoll, 2009*). The stratigraphic occurrence of *B. prima* in the Emanuel Creek type section between 131.5–152.8 m places the species between the *Paroistodus parallelus* and *Prioniodus oepiki–Serratognathus bilobatus* conodont zones (*Zhen & Nicoll, 2009*), equivalent to the *Paroistodus proteus* Zone in Baltica (*Percival, Zhen & Normore, 2023*, and references therein). This suggests a late Lancefieldian (La3) to early Bendigonian (Be1) age (*VandenBerg, 2018*; *Zhen et al., 2021*), corresponding with the early Floian in the global Ordovician timescale (*Bergström et al., 2009* and references therein).
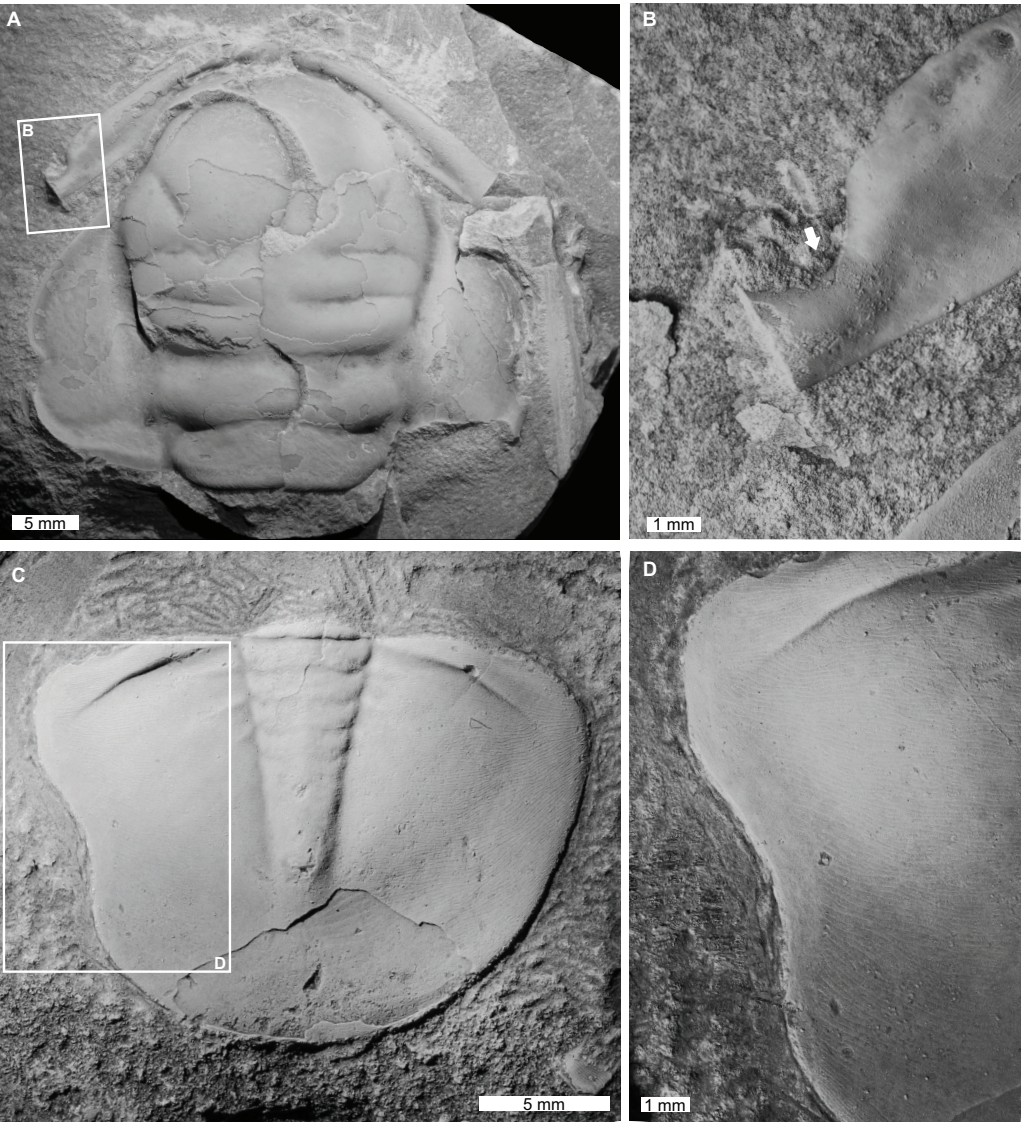

**Figure 3 Injured trilobites from the Cambrian Devoncourt Limestone and Ordovician Emanuel Formation.** (A and B) *Centropleura phoenix Öpik, 1961*, CPC 3494, paratype, from the Cambrian (Miaolingian, Guzhangian) Devoncourt Limestone, Queensland. (A) Entire specimen. (B) Close up of malformed region showing U-shaped indentation on anterior cranidial border (white arrow). (C and D) *Burminresia prima* Laurie & Shergold, 1996b, CPC 31981, paratype from the Ordovician (early Floian) Emanuel Formation, Western Australia. (C) Entire specimen. (D) Close up of malformed region showing U-shaped indentation on left side and a lack of exoskeletal ornament surrounding the injury.

Walker Volcanics—Paratype CPC 18440 of *Coronocephalus urbis Strusz, 1980* (Fig. 4) was collected from "locality 101" within the upper Walker Volcanics in the Lachlan Orogeny, at a site on top of a scarp on the bank of Molonglo River (approximately 35°16′ 30.0″ S 149°01′08.9″ E), Canberra, Australian Capital Territory (*Strusz, 1980*). Here the unit is a 5-m-thick sequence of siltstone and limestone underlain by thick beds of felsic volcanics and overlain by an eroded cover of calcareous and tuffaceous shale of the same unit (*Strusz, 1980*; *Abell, 1982*). The fossiliferous sediments were likely deposited in a quiet

marine setting during quiescent periods in-between pyroclastic flows (possibly as part of an island arc setting; *Abell, 1991*; *Pickett et al., 2000*). Unfortunately, no age-diagnostic faunas are known from "locality 101". However, the Walker Volcanics is generally thought to stratigraphically underlie the Silverdale Formation and Laidlaw Volcanics regionally (*Abell, 1982*, *1991*), which *Simpson (1995)* argued were Homerian (mid-Silurian) in age. An older age for the Walker Volcanics was suggested more recently with the description of the spiriferid brachiopod *Hedeina oepiki Strusz, 2010*, as it also occurs in the Canberra Formation. The latter unit is more reliably dated by the graptolite *Monograptus flemingii* (*Salter, 1852*), indicating a maximum age within the *Pristiograptus dubius* to *Cyrtograptus lundgreni* zones (*Perrier et al., 2015*). Therefore, the fossiliferous horizons within the Walker Volcanics were likely deposited during early Wenlock (Sheinwoodian) times in the Silurian (*Perrier et al., 2015*; *Bicknell & Smith, 2021*).

## Terminology

**Injury**: Exoskeletal breakage resulting from a predatory attack, scavenging, moulting complications, or mechanical impacts when the animal was alive (*Bicknell et al., 2022a*). Injuries are commonly L-, U-, V-, or W-shaped indentations (*Babcock, 1993*; *Bicknell & Pates, 2019*; *Bicknell et al., 2022a*), or a 'single spine injury' (SSI; *sensu Pates & Bicknell, 2019*; *Bicknell & Pates, 2020*; *Bicknell et al., 2022a*), and often show cicatrisation and/or segment regeneration. Rarely, exoskeletal areas can recover abnormally, resulting in fusion of exoskeletal regions and possible lack of segment expression (*Conway Morris & Jenkins, 1985*; *Owen, 1985*; *Bicknell et al., 2022a*, *2023*).

**Malformation**: Evidence for injuries, teratologies, or pathologies; the last of these are not documented here.

**Teratology**: Expressions of inferred genetic or developmental malfunctions (*Owen, 1985*). Examples include: additional, atypically absent, or offset segments and spines; fusion or bifurcation of segments; and abnormally developed morphologies, such as asymmetrical exoskeletal regions (*Strusz, 1980*; *Howells, 1982*; *Owen, 1985*; *Bicknell & Smith, 2021*, *2022*).

## RESULTS

### Malformations

*Galahetes fulcrosus Öpik, 1975*, CPC 10348, holotype, Cambrian (Miaolingian, Wuliuan), Beetle Creek Formation, Queensland, Australia (Figs. 1A and 1B): Mostly complete exoskeleton consisting of a partial cranidium, thorax, and partial pygidium, with a total sagittal body length of 27.8 mm. Thoracic pleurae 12 and 13 on the right pleural lobe are malformed. These pleurae are fused on the inner portions between the axial rings and the fulcra. The anterior boundary of the 13th pleura is observed, but poorly defined compared to other thoracic segments (Fig. 1B, black arrows). The 13th right pleura is truncated distally (relative to the 12th pleura) with a rounded tip. The 11th and 12th axial rings on the thorax also appear to be medially fused.

*Xystridura altera Öpik, 1975*, CPC 10407, Cambrian (Miaolingian, Wuliuan to Drumian), Arthur Creek Formation, Northern Territory, Australia (Figs. 1C and 1D): Incomplete exoskeleton consisting of a partial thorax and pygidium that are slightly

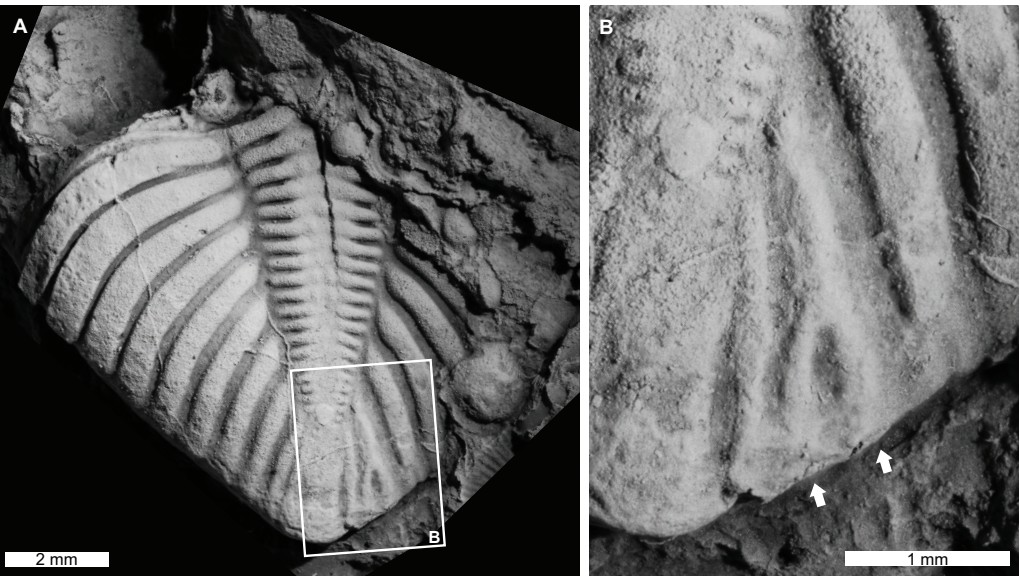

**Figure 4 Teratological pygidium of *Coronocephalus urbis* Strusz, 1980 from the Walker Volcanics, Silurian (Wenlock, Sheinwoodian or early Homerian), Australian Capital Territory.** (A and B) Latex cast of CPC 18440. (A) Entire specimen. (B) Close up showing bifurcation of pleural rib (white arrows).

disarticulated. The specimen has a total sagittal length of 29.5 mm, including displacement. The pygidium has a large W-shaped indentation on the right side. The indentation is 13 mm long and truncates the distal pygidial posterior marginal spine. This spine is showing evidence of recovery (Fig. 1D, black arrow).

*Papyriaspis lanceola Whitehouse, 1939*, CPC 44539, Cambrian (Miaolingian, Drumian), V-Creek Limestone, Queensland (Figs. 2A and 2B): Partial exoskeleton consisting of an incomplete cranidium, thorax, and pygidium that has a total sagittal body length of 43.9 mm. Thoracic pleurae 11–14 on the left pleural lobe show an L-shaped indentation that is 5.9 mm long and truncates the left pleural lobe by 1.7 mm when compared to the right side of the thorax. The malformed pleurae show rounding, wrinkling, and pinching at the tips (Fig. 2B). This malformation may have extended anteriorly, but this cannot be confirmed as CPC 44539 is broken along the rock edge in this exoskeletal region.

*Dolicholeptus ansatus Öpik, 1982*, CPC 18897, paratype, Cambrian (Miaolingian, Drumian), V-Creek Limestone, Queensland, Australia (Fig. 2D): Pygidium that is 4.1 mm long (sag.) and 7.5 mm wide (tr.). *Öpik (1982*, p. 45) had proposed that this pygidium is "abnormal", noting that it shows evidence for part of an earlier moult retained on the posteriormost region of the axial lobe (Fig. 2D, white arrows). However, re-examination of the specimen shows that the so-called abnormality likely represents partial collapse of the axis during compaction.

*Dolicholeptus licticallis Öpik, 1982*, CPC 18907, paratype, Cambrian (Miaolingian, Drumian), V-Creek Limestone, Queensland, Australia (Fig. 2E): Pygidium that is 4.4 mm long (sag.) and 8.8 mm wide (tr.). The pygidium is asymmetrical, with the posterior region of the axial lobe deflected to the right, the second axial ring is shorter (exsag.) on the right

side than the left side, and the right posterior margin is slightly thicker compared to the left side.

*Centropleura phoenix Öpik, 1961*, CPC 3494, paratype, Cambrian (Miaolingian, Guzhangian), Devoncourt Limestone, Australia (Figs. 3A and 3B): Partial cranidium that has a sagittal length of 34.0 mm. The specimen displays a U-shaped indentation on the left corner of the anterior border (Fig. 3B), deflecting the anterior border furrow adjacent to it. This indentation is 2.7 mm long, extending 1.4 mm inwards towards the anterior border furrow and is not observed on the right side of the cranidium.

*Burminresia prima Laurie & Shergold, 1996b*, CPC 31981, paratype, Emanuel Formation, Ordovician (early Floian), Western Australia, Australia (Figs. 3C and 3D): Pygidium that is 15.2 mm long (sag.) and 20.1 mm wide (tr.). The pygidium has a large U-shaped indentation on the left side that is 6.9 mm long and extends 0.6 mm inwards from the posterior margin (Fig. 3D). A narrow (*ca.* 1 mm wide) strip of the dorsal surface adjacent to the U-shaped indentation appears to be smooth, in contrast to the fine terrace ridges that extend to the pygidial margin on the right side.

*Coronocephalus urbis Strusz, 1980*, CPC 18440, paratype, Walker Volcanics, Silurian (Wenlock, Sheinwoodian or early Homerian), Australian Central Territory, Australia (Figs. 4A and 4B): Partial pygidium that is 8.2 mm long (sag.) and 7.6 mm wide (tr.). The second to last pygidial rib on the right side bifurcates 1.3 mm from the axial furrow, forming a V-shaped abnormality (Fig. 4B, white arrows).

### Carcass

Specimen CPC 44540 is an incomplete exoskeleton of *Asthenopsis* sp. consisting of a partial thorax and pygidium that has a sagittal length of 29.4 mm (Fig. 2C). Substantial sections of the axial and right pleural lobes in the posterior region of the thorax are missing and broken thoracic segments of the left pleural lobe have been partly displaced. This specimen contrasts the nicely articulated exoskeletons of similar-sized trilobites from the V-Creek Limestone (*Öpik, 1961*, *1982*; *Jell, 1978*, *1989*).

## DISCUSSION

### Injuries

Observed malformations on *Galahetes fulcrosus*, *Xystridura altera*, *Papyriaspis lanceola*, *Burminresia prima*, and *Centropleura phoenix* (Figs. 1A–1D, 2A, 2B and 3A–3D) are comparable to previously documented examples of injured Cambrian (*Rudkin, 1979*; *Jell, 1989*; *Babcock, 1993*, *2007*; *Pates et al., 2017*; *Bicknell & Pates, 2020*; *Zong, 2021a*, *2022b*; *Bicknell et al., 2022c*) and Ordovician (*Ludvigsen, 1977*; *Babcock, 2007*; *Zong, 2021b*; *Bicknell et al., 2022b*, *2022c*) trilobites. These similarities include the malformation outlines (W-, U-, and L-shaped indentations) and their prevalence on the trunk (*i.e.*, thorax and/or pygidium). Given these parallels, we suggest these examples likely record evidence for failed predation or complications due to moulting.

The *Galahetes fulcrosus* specimen (Figs. 1A and 1B) illustrates a combination of malformations from an injury, including fused posterior thoracic segments and a truncated right pleura on the 13[th] segment. These malformations are comparable to

injuries documented on a heavily malformed *Estaingia bilobata Pocock, 1964* from the Emu Bay Shale (Cambrian Stage 2, Series 4) (*Bicknell et al., 2023*, fig. 2e, f). The disruption of thoracic segments indicates that the individual may have been attacked shortly after moulting, and the attack was severe enough to impact the axial region of the exoskeleton. Despite the axial lobe housing musculature, the digestive system, and other important soft tissues (*Whittington, 1993*; *Lerosey-Aubril, Hegna & Olive, 2011*; *Lerosey-Aubril et al., 2012*; *Wang et al., 2018*), this individual survived the predation attempt. The fusion of the posterior regions demonstrates that the abnormal morphologies that arose through injury recovery were propagated through subsequent moulting events (*Conway Morris & Jenkins, 1985*). Furthermore, the near bilateral expression of the exoskeleton suggests that the specimen may have experienced multiple moults after the attack (*McNamara & Tuura, 2011*; *Pates et al., 2017*; *Zong & Bicknell, 2022*).

The injured *Xystridura altera* with a W-shaped indentation on its pygidial margin is comparable to injuries on other trilobites that are attributed to predation (*Owen, 1985*; *Bicknell et al., 2022a*). Importantly, the injury has evidence of a regenerating pygidial spine (Fig. 1D), demonstrating that the individual survived the attack and had subsequently moulted. Although we can only assess one moult stage, it seems likely that the spine recovery process was similar to the repair sequence for trilobite thoracic pleural spines (*Pates et al., 2017*).

Two of the trilobites from the V-Creek Limestone documented here exhibit clear signs of damage caused by predators and/or scavengers (Figs. 2A–2C). The specimen of *Papyriaspis lanceola* has an L-shaped injury to the thorax, with pleural tips showing deformation and recovery (Figs. 2A and 2B). This type of malformation is similar to rare predation injuries on *Redlichia takooensis Lu, 1950* and *Redlichia rex Holmes, Paterson & García-Bellido, 2019* from the Emu Bay Shale (*Bicknell et al., 2022a*). Another previously documented specimen of *P. lanceola* from the V-Creek Limestone with a large U-shaped indentation on the thorax (*Jell, 1989*, fig. 6) was attributed to failed predation or accidental tearing of the poorly reinforced exoskeleton through mating (*Jell, 1989*). As L- and U-shaped injuries showing signs of recovery are considered evidence for failed predation (*Rudkin, 1979*; *Bicknell & Paterson, 2018*; *Bicknell et al., 2022a*), we propose that these *P. lanceola* individuals were attacked by predators, but subsequently survived. Conversely, the highly disrupted exoskeleton of *Asthenopsis* sp. (Fig. 2C) is suggestive of either a lethal attack by a durophagous predator or post-mortem scavenging (*Bicknell & Paterson, 2018*; *Bicknell et al., 2022a, 2023*).

Ordovician asaphid pygidia have been documented with U-, V-, and W-shaped indentations that have been attributed to failed predation (*Šnajdr, 1979*; *Rudkin, 1985*; *Babcock, 1993*; *Bicknell et al., 2022b*; *Bicknell & Kimmig, 2023*; *Bicknell & Smith, 2023*) or moulting complications (*Wandås, 1984*; *Bicknell & Kimmig, 2023*). In the case of the injured *Burminresia prima* (Fig. 3C), the cause is ambiguous, but the U-shaped indentation across almost half of the pygidium is suggestive of failed predation. Also, the lack of fine terrace ridges immediately adjacent to the injury shows that some exoskeletal morphology—in this case, surface ornamentation—could not be regenerated after a traumatic event.

Cephalic injuries are rare in Cambrian trilobites, a record that contrasts evidence from younger Paleozoic deposits (see *Resser & Howell, 1938*; *Alpert & Moore, 1975*; *Cowie & McNamara, 1978*; *Owen, 1985*; *Skinner, 2004*; *Fatka, Budil & Grigar, 2015*; *Bicknell & Paterson, 2018*; *Bicknell, Pates & Botton, 2018*; *Smith, Paterson & Brock, 2018*). Due to this rarity, the *Centrapleura phoenix* specimen displaying a U-shaped injury in the cephalic region (Figs. 3A and 3B) is important. As the cephalon housed vital organs, major damage to this region was likely fatal (*Babcock, 1993*; *Whittington, 1997*; *Bicknell & Paterson, 2018*). However, given the small size of the injury in this specimen, it was likely to be non-lethal and rounding of the indented margin demonstrates the individual moulted at least once. It is possible that the indentation reflects failed predation or accidental trauma during a soft-shelled stage (*Rudkin, 1985*).

## Teratologies

While asymmetrical pygidia are often associated with abnormal recovery from injuries (*Šnajdr, 1981a*; *Owen, 1985*), teratological explanations have also been presented (*Lee, Choi & Pratt, 2001*; *Kandemir & Lerosey-Aubril, 2011*). We propose that the *Dolicholeptus licticallis* pygidium (Fig. 2E) records a genetic malfunction of the second axial ring. Indeed, the axial ring asymmetry is comparable to abnormal rings in *Sceptaspis lincolnensis* (*Branson, 1909*; *Rudkin, 1985*) and *Ditomopyge*? sp. (*Kandemir & Lerosey-Aubril, 2011*) that were ascribed to developmental malfunctions. The thicker pygidial margin on the right side also requires explanation. We suggest that either the specimen was injured and recovered with a thicker border, or produced a thicker border associated with the asymmetry. Either possibility is plausible, representing the complex interplay and overlap between teratologies and possible injuries (*Owen, 1985*).

The *Coronocephalus urbis* pygidium with a bifurcated rib (Figs. 4A and 4B) is similar to other examples of abnormally divergent ribs (*Šnajdr, 1981a*, *1981b*; *Owen, 1985*; *Nielsen & Nielsen, 2017*; *Bicknell & Smith, 2021*). These structures are generally associated with genetic malfunctions that propagated through subsequent moults (*Owen, 1985*; *Bicknell & Smith, 2021*). It is possible that bifurcated ribs can reflect the teratological recovery of a minute injury (*Nielsen & Nielsen, 2017*). However, there is no exoskeletal deformation around the teratology to support this possibility. The fossil record of bifurcated ribs on pygidial regions also suggests that this region of the exoskeleton may have been more prone to genetic or developmental complications.

## Abundance of malformed specimens and future work

Determining the relative abundance of malformed trilobite specimens at the population level is often complicated by the small sample sizes for most species, especially in fossil repositories such as museum collections. The Beetle Creek Formation and V-Creek Limestone, in particular, are well-sampled formations (*Chapman, 1929*; *Whitehouse, 1936*, *1939*; *Öpik, Carter & Noakes, 1959*; *Shergold, 1969*; *Öpik, 1970a*, *1970b*, *1975*, *1979*, *1982*; *Hill, Playford & Woods, 1971*; *Jell, 1975*, *1978*; *Paterson, 2005*) and Geoscience Australia has good representative samples, permitting a possible assessment of relative abundance of trilobite malformations. However, as only one malformed Beetle Creek and four V-Creek

trilobites have been documented, there are limitations on what conclusions may be drawn from the available samples, especially when specimens come from different localities and stratigraphic levels. We propose that detailed field-based studies of specimens from these formations should be undertaken to not only uncover new examples of malformations, but also permit quantitative assessments of malformation abundances and their palaeobiological implications.

## CONCLUSIONS

Re-examining trilobites in the Commonwealth Palaeontological Collection and the associated Geoscience Australia bulk fossil collection has permitted us to study malformations previously considered anecdotal points of interest, as well as record new examples of malformations. Documentation of malformed specimens of *Asthenopsis*, *Burminresia, Centropleura, Coronocephalus, Dolicholeptus, Galahetes, Papyriaspis*, and *Xystridura* has allowed us to present new insight into the palaeobiology of Cambrian, Ordovician, and Silurian trilobites from Australia. In particular, we illustrate evidence for injuries, teratologies, and possible records of post-mortem scavenging. This highlights the need for these deposits to be examined in more detail to uncover new malformed specimens and determine their relative abundance at the population level.

## ACKNOWLEDGEMENTS

Thanks to Natalie Schroeder for assistance with the CPC and bulk collection at Geoscience Australia. We also thank Olev Vinn and Morten Nielsen for their reviews and Kenneth De Baets for editorial support.

### Funding

This research was funded by an Australian Research Council Discovery Project grant (DP200102005 to John R. Paterson), a University of New England Postdoctoral Fellowship (to Russell D. C. Bicknell) and a MAT Program Postdoctoral Fellowship (to Russell D. C. Bicknell). The funders had no role in study design, data collection and analysis, decision to publish, or preparation of the manuscript.

### Grant Disclosures

The following grant information was disclosed by the authors:
Australian Research Council Discovery: DP200102005.
University of New England Postdoctoral Fellowship.
MAT Program Postdoctoral Fellowship.

### Competing Interests

The authors declare that they have no competing interests.

## Author Contributions

- Russell D. C. Bicknell conceived and designed the experiments, performed the experiments, analyzed the data, prepared figures and/or tables, authored or reviewed drafts of the article, and approved the final draft.
- Patrick M. Smith performed the experiments, authored or reviewed drafts of the article, and approved the final draft.
- John R. Paterson conceived and designed the experiments, prepared figures and/or tables, authored or reviewed drafts of the article, and approved the final draft.

## Data Availability

All specimens are housed within the Commonwealth Palaeontological Collection, Geoscience Australia, Canberra, Australian Central Territory, Australia. Specimens were assigned the following CPC numbers: CPC 3494, CPC 10348, CPC 10407, CPC 18440, CPC 18897, CPC 18907, CPC 31981, CPC 44539, and CPC 44540.

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
