# Peer review of "Malformed trilobites from the Cambrian, Ordovician, and Silurian of Australia"

_PeerJ, doi:10.7717/peerj.16634_

## Round 0.1 · original submission · Minor Revisions

Congratulations on the description and thorough characterizations of eight malformed trilobite and one disarticulated specimen. I apologize for the delayed decision but was waiting for an additional review to come in which unfortunately did not come in. I agree with the reviewers that the manuscript is in a good state, but some minor but crucial points need be addressed before publication. The main points being:

Abstract: I agree with reviewer 2 that it would be beneficial and more informative to list the type of structure and its interpretation in the abstract

Introduction: I agree with reviewer 2 that the introduction would benefit from briefly presenting the different types of malformations within the manuscript and their characteristics (e.g., genetic abnormalities – teratologies). I agree with reviewer 2 that the manuscript would benefit from a more explicit description of the motivation/importance of studying malformations beyond predator-prey relationships.

Prevalence: As you had first-hand access to the collections and publications, it would be crucial to mention explicitly the size sample of species where specimens showed malformations which are available in similar preservation and from the same locality/stratigraphic unit. This would be helpful to understand how common such malformations are and would be crucial to reproduce/compare such studies at other localities as well as help to constrain causes behind such pathologies. It would also better support the rationale for including detailed stratigraphic description (a point raised by reviewer 2). I know this might be unavailable for some findings, but as malformed specimens are usually mentioned on the side, more specimens for each species than those were available. As you had/have first-hand access to the specimens/collections and publications, you would be uniquely suited to make decision on which specimens are similarly and suitable (completely enough) preserved to support their absence.

Results: Please also describe your own interpretations of CP18907 (in case there are any) in the results for the sake of completeness (compare reviewer 2)

Conclusions: I agree with reviewer 3 that not having an explicit conclusion section is a missed opportunity.

Figure order: In PeerJ figures are usually published in the order of first mention in text so there seems to be discrepancy with the carcass on Figure 2C (compare reviewer 2). However, if this is not an issue for you, then it could also be kept like this.

Please make sure to address these points as well as all others pointed raised by reviewers and myself (see annotated pdf).

I look forward to receiving your revised manuscript.

·

Basic reporting

This is very well written and interesting paper on the malformed trilobites from the Cambrian, Ordovician, and Silurian of Australia. Authors have used clear, unambiguous, and professional English throughout the MS. Introduction is well written and shows the background of the research. Literature is well referenced and relevant. Structure of the MS conforms to PeerJ standards. All figures are relevant, high quality, well labelled and described in the text. The research hypothesis are fine.

Experimental design

Article represents an original primary research within Scope of PEERJ. All research question are well defined, relevant and highly meaningful. It is stated how the research fills an identified knowledge gap.
Rigorous investigation has been performed to a high technical standard. Methods are described with sufficient detail and allow research to be replicated.

Validity of the findings

Authors have provided all underlying data which are robust and
controlled. All conclusions are well stated, linked to original research question and limited to supporting results.

Additional comments

This excellent paper can be published in its current form. I very much liked reading of your MS.

Congratulations!

Olev Vinn

·

Basic reporting

The objective of the manuscript is to review a national museum collection of Australian fossils for trilobite malformations. The review was motivated by the presence of multiple malformed specimens in publications from the 1960s-1980s, some of which were written by the highly influential trilobite palaeontologist Armin Öpik. The manuscript describes eight malformations and a disarticulated specimen and discusses their possible causes.

The manuscript is well-written and generally easy to read. The figures present the evidence well.

The structure is could in some places benefit from a revision in some places to improve the clarity and flow:

• The abstract presents its content as “Most of the malformed specimens reflect injuries from either failed predation or complications during moulting, and a mangled carcass is ascribed to either successful predation or post-mortem scavenging. We also uncover examples of teratologies and abnormal recovery from traumatic incidents, highlighting the range of morphologies that can be derived from these processes“(line 26-30). Perhaps it would be a bit more informative if it instead listed the type of structure and its interpretation (fx as: “Four indentions in W-, U-, and L-shapes are interpreted as injuries from predation attempts”, or something.)

• The introduction only mentions the different types of malformations. It would benefit from briefly presenting the different types of malformations within the manuscript and perhaps their characteristics. For example, the introduction mentions “genetic abnormalities” (line 35) but does not relate it to teratologies.

• The geological context is detailed for each locality and provides a good locality description for resampling localities. It also provides their depositional environments and biozones. The latter is a bit long, especially as it seems tangential to the manuscript.

• The discussions first paragraph (line 261-267) says that “Observed malformations […] are comparable to previously documented examples of injured […] trilobites.” I suggest elaborating how they are comparable by stating their shared characteristics.

• CP18907 (Fig. 2D) is not described in Results (line 225-228, but only referred to by its previous interpretation by Öpik (1982). Its discussion (line 329-332) does therefore not have a basis in the text.

• The figures are not presented in the texts’ order. Carcass (line 252) is described as the last specimen but shown on Figure 2C.

Experimental design

The motivation for studying malformations is not clearly stated and the manuscript does therefore not currently state an addressed question, other than reviewing a fossil collection. The introduction says that “Specimens of extinct animals displaying malformations present important, and often 35 unique insight into predation, pathological growths, and genetic abnormalities in the fossil record” but it would be nice with a short explanation on why these insights are important. Bicknell and Paterson have already done well to explain that these studies are helpful for understanding predator-prey relationships through time in Bicknell et al. 2022 (https://doi.org/10.1016/j.palaeo.2022.110877).

Descriptions of malformations are concise with proper terminology and well-connected to figures (except for CP18907, Fig. 2D; see ‘Basic Reporting’).

Material and methods are clearly stated and in appropriate detail.

Validity of the findings

Conclusions are not included as a section in the manuscript, which is a missed opportunity to emphasize the key point(s) and why it may be important.

All data presented and discussed is presented in the figures to the required standard.

Additional comments

These are minor comments for clarity referred to by their line of occurrence.

Line 208: “The 13th pleura is truncated…” Truncated distally?
Lines 252-258: Two alternative causes that might be worth considering if you haven’t already: 1) Can preparation be excluded as a cause for the disarticulation on the right side? It appears, from the photo, to have been heavily prepared. 2) It appears to be partially flexed. Could the specimen perhaps represent an incomplete moulting?
Line 270: “The extent of these malformations indicates that this was the result of failed predation” Please explain how.
Line 280: “… comparable to injuries attributed to predation on other trilobites” Probably along other predators too, there could have been many artiopods around.
Line 303: “…but the shape and extent of the injury is suggestive of failed predation.” It would help to explain which shape and extent is required for this interpretation (not that I think you are wrong).


I hope the authors will find my comments helpful to improve the manuscript. I believe these findings, despite having been treated as mere curiosities in older taxonomic publications, are important to detect overall patterns in fossil (trilobite) malformations that can help us better understand ecological relationships and genetic versatility of extinct organisms.

Good luck on the manuscript.

Best wishes,
Morten Lunde Nielsen

---

## Round 0.2 · accepted · Accept

Thank you for considering and addressing all suggestions. Particularly the addition of the new section on terminology and discussion on relative abundance make the manuscript even easier to follow and of broader relevance for the community. I look forward to seeing this published.